# Combined Transcriptomic and Metabolomic Analyses of Defense Mechanisms against Phytoplasma Infection in *Camptotheca acuminata* Decne

Kai Qiao, Weiyi Huang, Xuemei Li, Jiahui Liang and Hong Cai *

Phytopathology Laboratory of Yunnan Province, College of Plant Protection, Yunnan Agricultural University, Kunming 650201, China; 13359239716@163.com (K.Q.); 18213484636@163.com (W.H.); lxm2765489@163.com (X.L.); liangjiahui020502@163.com (J.L.)
* Correspondence: caihong0623@126.com

**Abstract:** *Camptotheca acuminata* Witches'-broom disease (CaWB) is the most destructive disease affecting *C. acuminata* in China. Previous studies on CaWB have failed to clarify the incidence pattern in *C. acuminata* after infection with phytoplasma. The time interval between phytoplasma infection of *C. acuminata* and the onset of Witches'-broom symptoms in *C. acuminata* was very long. *C. acuminata* inoculated with CaWB showed leaf margin scorching symptoms at 4 weeks in inoculated leaves. At 16 weeks after infection (WAI), old leaves were shed, while new leaves showed a mild leaf margin scorch; at 28 WAI, typical symptoms appeared. Transcriptomic and metabolomic analyses of the three sampling periods revealed 194 differentially expressed genes, mainly enriched in MAPK signaling, plant–pathogen interaction, phenylpropanoid biosynthesis, starch and phenylpropanoid biosynthesis, and phenylpropanoid biosynthesis pathways. The expression of calcium-dependent protein kinase (CDPK), β Ketoacyl-CoA Synthase1/10 (KCS1/10), and WRKY22/29 genes in the plant–pathogen interaction pathway significantly increased, indicating that they may be key genes in the CaWB phytoplasma-mediated maintenance of ROS homeostasis. Moreover, isochlorogenic acid B, atractylenolide II, and 3-methoxybenzoic acid were found, which might serve as signaling or functional substances in the defense response. Our results provide novel insights into the pathogenesis of CaWB and the defense response of *C. acuminata* under the influence of phytoplasma. Additionally, we identified potential candidate genes related to the defense response of *C. acuminata*, laying the foundation for further research.

**Keywords:** transcriptome; metabolome; phytoplasma; *Camptotheca acuminata* Decne; defensive response

## 1. Introduction

*Camptotheca acuminata* Decne. *Nyssaceae*, *Camptotheca* Decne, a deciduous tree native to China primarily found in the Yangtze River valley [1,2], is valued for its medicinal properties and high levels of camptothecin, a compound with anticancer properties [3,4]. *C. acuminata* Witches'-broom phytoplasma (CaWB) is the most destructive disease affecting *C. acuminata* in China. CaWB is caused by phytoplasma (16SrXXXII) [5] and transmitted by *Empoasca (Matsumurasca)* paraparvipenis [6]. The typical symptoms of CaWB phytoplasma infection include *C. acuminata*-topped witches' brooms and browning.

Phytoplasmas are prokaryotic pathogens that lack cell walls, parasitize plant phloem and insects, and cannot be artificially cultivated. Phytoplasma can freely move through vascular bundles in plants and cause diseases such as Witches'-broom disease, stem flattening, leaflet abnormalities, and flower greening in various economically important crops [7,8]. Most of these symptoms can be reflected by specific changes in the transcriptome and metabolome of infected plants compared with that of healthy plants. Common plant responses to phytoplasma infection include downregulation of genes associated with photosynthesis and corresponding changes in protein levels, alterations in carbohydrates

at the polyomics level, increase in the levels of plant secondary metabolites, activation of flavonoid production genes, and alterations in the expression of plant defensive response genes [9–13]. miRNAs have also been found to reduce plant immune responses in the defense against phytoplasma infections through post-transcriptional gene regulation. Long-stranded non-coding RNAs may regulate reactive oxygen species (ROS) and hypersensitivity responses [14,15].

Phytoplasma lacks genes for many biometabolic pathways, including the pentose phosphate pathway; therefore, they are highly dependent on the host to supply pathway-deficient products and metabolites [16,17]. Plants infected with phytoplasma tend to exhibit elevated glucose, sucrose, polyphenol, and succinate levels [18]. ATP synthesis in phytoplasmas is highly reliant on the glycolytic pathway [16]. Therefore, a high-carbon environment is conducive to phytoplasma proliferation [16] and may lead to more severe symptoms in diseased plants [19]. Plants have developed a complex set of processes to improve their survival in the fight against pathogens, including the production of multiple secondary metabolites to resist the damage caused by pathogen inoculation [20,21]. In studies on phytoplasma-infected date palms [22], paulownia [12], coconut palms, and grapevines [23], lignin, phenylpropanoid, and flavonoid synthase genes were found to be upregulated.

The mechanisms by which host plants respond to phytoplasma infections are complex. The time interval between phytoplasma infections and the onset of Witches'-broom symptoms in *C. acuminata* is very long. As time progresses, diseased *C. acuminata* begins to manifest a series of symptoms (Figure 1). The physiological and molecular mechanisms underlying phytoplasma infection remain nebulous. Therefore, in this study, we analyzed the transcriptome and metabolite profiles of *C. acuminata* at different time points following inoculation with phytoplasma and explored its disease incidence pattern. This study provides novel insights into the incidence pattern in *C. acuminata* after being infected with phytoplasma, facilitating future research on the mechanisms underlying Phytophthora-plant interactions.

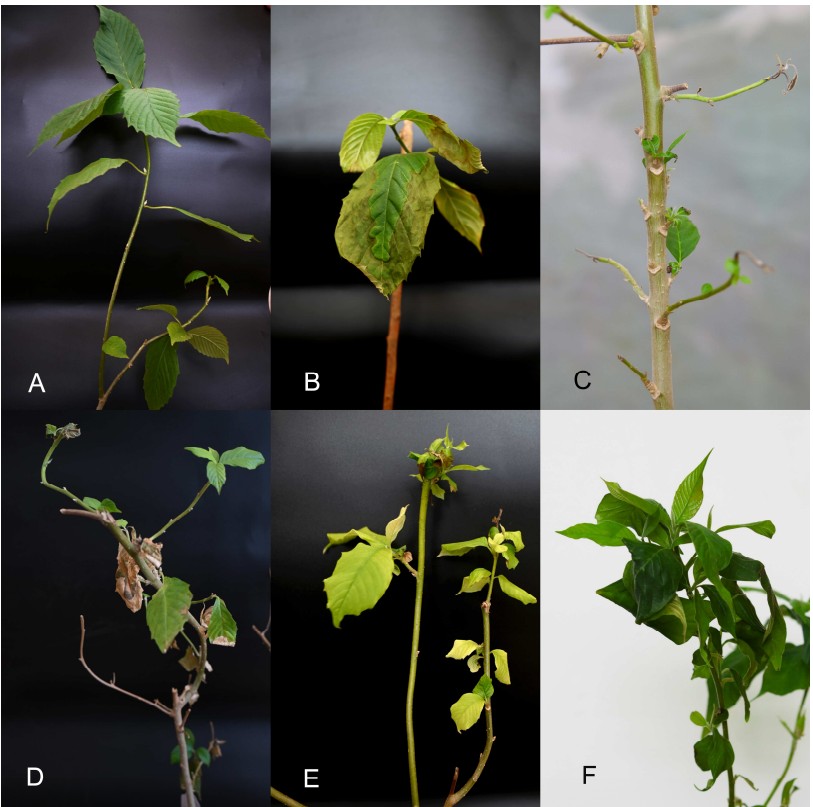

**Figure 1.** Symptoms of *Camptotheca acuminata* at different stages of *C. acuminata* Witches'-broom phytoplasma infection. (**A**) 0 WAI; (**B**) 4 WAI; (**C**) 10 WAI; (**D**) 16 WAI; (**E**) 20 WAI; (**F**) 28 WAI. WAI, weeks after infection.

## 2. Materials and Methods

Three-year-old CaWB-infected and healthy *C. acuminata* were cultivated in a greenhouse with a screen mesh on the fifth floor of the School of Plant Protection, Yunnan Agricultural University, China. Healthy vector insects *Empoasca* (*Matsumurasca*) paraparvipenis, Zhang and Liu, were co-cultivated with five pots of healthy *C. acuminata* in insect-rearing cages to maintain populations. Prior to the formal trials, 1 g of healthy *C. acuminata* terminal leaves was sampled, and 10 healthy third-instar *E.* (*Matsumurasca*) were captured and collected once a week, three times. DNA was extracted from the samples using an OMEGA E.Z.N.A.®SP Plant DNA Kit (D5511-02) and a micro Elute genomic DNA kit (D3096-1). Direct and nested PCR assays were carried out using the phytoplasma universal primer sets P1/P7 and R16F2n/R16R2, respectively, to ensure that the test material was not contaminated.

Fifty CaWB-free third instar nymphs of *E.* (*Matsumurasca*) were confined to each CaWB-infected plant and allowed to feed for a 3-day acquisition access period (AAP), followed by a 5-day latency period on healthy *C. acuminata*, and then transferred to healthy test plants for a 3-day inoculation access period (IAP). CaWB was transmitted to healthy *C. acuminata* using a leafhopper vector. The control and treatment groups were grown separately in identical screen mesh cages under the same cultivation conditions; defoliation occurred in the treatment group at 10 weeks after infection (WAI), at which point the control leaves were removed to retain the bud points and ensure the same growth as the treatment group.

### 2.1. DNA Extraction and PCR Analysis

To understand phytoplasma inoculation, *C. acuminata* leaves were collected weekly during the first month after inoculation and subsequently at 8, 16, 20, and 28 WAI. Additionally, CaWB-infected plants and IAP-stage *E.* (*Matsumurasca*) were used as positive controls, and healthy *C. acuminata* tissues were used as negative controls.

An OMEGA plant DNA extraction kit (D5511-02) and an OMEGA micro Elute genomic DNA kit (D3096-1) were used to extract total DNA from the tissue according to the manufacturers' instructions. The 16S rDNA gene of the phytoplasma was amplified via PCR using the DNA of the infected *C. acuminata* plants while employing the universal primer for phytoplasma designed by Lee et al. [24,25]. The primers used were (5'-3') P1 (AAGAGTTTGATCCTGGCTCAGGATT), P7 (CGTCCTTCATCGGCTCTT), R16F2n (GAAACGACTGCTAAGACT), and R16R2 (TGACGGGGGGTGTGTGTGTAAAA-CCCCCG). PCR amplification was performed using Phanta Max Super-Fidelity DNA polymerase and 2 Phanta buffer (Nanjing, China) to avoid introducing errors into the sequence. PCR products were detected via electrophoresis on a 1% agarose gel.

### 2.2. Sample Collection and Transcriptome Analysis

Based on the PCR results and observation of the symptoms of the plants after inoculation, the top leaf samples at three time points after inoculation and the leaf samples of healthy plants in the same period were collected for RNA extraction and transcriptome analysis. The three experimental periods were 4, 16, and 28 WAI. Each sample weighed approximately 1 g, and each replicate sample was obtained from three plants. Three groups of biological replicates were set up, frozen with liquid nitrogen, and stored at $-80\ ^\circ$C.

Total RNA was extracted from infected and healthy leaves using an RNA Pure Plant Kit (DP441; Tian Gen, Beijing, China). The quality of the extracted RNA was evaluated using the RNA Nano 6000 Assay Kit for the Bioanalyzer 2100 System (Agilent Technologies, Santa Clara, CA, USA). RNA samples with an $A_{260}/A_{280}$ ratio ranging from 1.8 to 2.0 were selected for transcriptome sequencing. Subsequently, libraries were prepared and sequenced on the Illumina NovaSeq 6000 platform (Illumina, San Diego, CA, USA).

To obtain clean data for the analysis, low-quality sequences were removed from both ends of the dataset using a threshold value of 30. Additionally, adapter contamination and reads with lengths < 60 bp were eliminated. The resulting reads were aligned to the *C. acuminata* genome (https://doi.org/10.6084/m9.figshare.12570599 (accessed on

14 April 2023)). The mapped results were then subjected to a BLAST search against the UniProtKB database (https://www.uniprot.org/uniprotkb (accessed on 14 April 2023)) to obtain annotation information. Differentially expressed genes (DEGs) were identified based on a fold change of at least 2 and a *p*-value of <0.05. To perform Gene Ontology (GO) enrichment analysis on the DEGs, the clusterProfiler R package (version 3.5.0, R Core Team, Vienna, Austria) was utilized, which corrects for any bias in gene length. Furthermore, the clusterProfiler R package was employed to assess the significant enrichment of DEGs in the Kyoto Encyclopedia of Genes and Genomes (KEGG) pathways [26–28].

*2.3. Metabolome Analysis*

2.3.1. Tissue Sample Metabolite Extraction

The top leaves of *C. acuminata* were collected at 4, 16, and 28 WAI, and leaf samples of healthy plants during the same period were used as controls (Section 2.2) for metabolite extraction and metabolome analysis. Each sample (100 mg) was ground and pulverized in an EP tube after flash freezing with liquid nitrogen; 500 μL of pre-cooled 80% methanol was added and thoroughly mixed using a vortex. The sample mixture was allowed to stand on ice for 5 min, followed by centrifugation at $15,000\times g$ and 4 °C for 20 min. A portion of the supernatant was diluted with LC–MS grade water to a final concentration of 53% and centrifuged at $15,000\times g$ and 4 °C for 20 min, and the supernatant was collected and injected into the LC–MS system for subsequent analysis [29].

2.3.2. HPLC–MS/MS Analysis

The LC–MS/MS analyses were conducted at Novogene Co., Ltd. (Beijing, China) using an ExionLC™ AD system (SCIEX) coupled with a QTRAP® 6500+ mass spectrometer (SCIEX). The samples were injected onto an Xselect HSS T3 column (2.1 × 150 mm, 2.5 μm) using a 20 min linear gradient at a flow rate of 0.4 mL/min for the positive/negative polarity mode. Eluents A (0.1% formic acid–water) and B (0.1% formic acid–acetonitrile) were used. The solvent gradient was set as follows: 2% B for 2 min, followed by a gradient from 2% to 100% B over 15 min, holding at 100% B for 17 min, returning to 2% B over 0.1 min, and finally holding at 2% B for 20 min [30].

The QTRAP® 6500+ mass spectrometer was operated in positive polarity mode with a curtain gas pressure of 35 psi, medium collision gas, an ion spray voltage of 5500 V, a temperature of 550 °C, and ion source gas of 1:60 and 2:60. In negative polarity mode, the QTRAP® 6500+ mass spectrometer was operated with a curtain gas pressure of 35 psi, medium collision gas, an ion spray voltage of −4500 V, a temperature of 550 °C, and ion source gas of 1:60 and 2:60.

2.3.3. Metabolite Identification and Quantification

Identification and quantification of the experimental samples were conducted using multiple reaction monitoring based on an in-house database developed by Novogene. Metabolite quantification was performed using Q3, whereas metabolite identification utilized Q1, Q3, retention times, declustering potentials, and collision energy. The data files generated using HPLC–MS/MS were processed with SCIEX OS Version 1.4 to integrate and correct the peak. The processing parameters were set as follows: minimum peak height of 500, signal-to-noise ratio of 5, and Gaussian smoothing width of 1. The area of each peak represents the relative contents of the corresponding substances.

The KEGG, Human Metabolome Database (HMDB), and Lipidmaps databases were used for metabolite annotation. Principal component analysis (PCA) and partial least squares discriminant analysis (PLS-DA) were performed using MetaX [30], a comprehensive software for processing metabolomic data. Univariate analysis (*t*-test) was used to calculate statistical significance (*p*-value). Metabolites with variable importance in projection score > 1, *p*-value < 0.05, and fold change of at least 2 or <0.5 were considered differentially abundant metabolites. Volcano plots were generated using ggplot2 in R to filter the metabolites of interest based on log2-fold change and -log10 *p*-value. Clustering heatmaps

were created using the Pheatmap package in R, with data normalized using z-scores of the intensity areas of differentially abundant metabolites. The correlation between differentially abundant metabolites was analyzed using the cor function in R using the Pearson correlation method. Statistically significant correlations between differentially abundant metabolites were calculated using the correlation test function in R. $p < 0.05$ was considered statistically significant; correlation plots were generated using the corrplot package in R.

The functions of the identified metabolites and their involvement in metabolic pathways were investigated using the KEGG database. Metabolic pathway enrichment analysis of differentially abundant metabolites was performed. Metabolic pathways were considered enriched when the x/n ratio was greater than y/N and significantly enriched when the $p$-value was <0.05.

### 2.4. Quantitative Real-Time PCR Analysis

We selected 10 genes associated with changes in secondary metabolites from the transcriptome data to validate the sequencing results. The CaUBC gene was used as an internal reference, and qPCR primers were designed using Primer 5.0 (Premier Biosoft International, Palo Alto, CA, USA). qPCR primers were used with HiScript II Total RNA, which was reverse transcribed using a HiScript II Reverse Transcription Kit (Vazyme, R223), and cDNA was diluted with 80 μL of RNase-free water. Quantitative real-time PCR was performed using the SYBR Green I chimeric fluorescence method and ChamQ SYBR qPCR Master Mix (Vazyme, R223). The reaction system was configured using ChamQ SYBR qPCR Master Mix (Vazyme, Q311) and a QuantStudio™ 5 Real-Time PCR System (Applied Biosystems). Amplifications were performed at 95 °C for 30 s, followed by 40 cycles of denaturation at 95 °C for 10 s, annealing at 60 °C for 30 s, and a final dissociation step at 95 °C for 15 s, 60 °C for 1 min, and 95 °C for 15 s. Three biological replicates were set up, and each reaction was repeated three times. The data were processed using the ΔΔCT method [31] and compared with the transcriptome expression values (FPKM) of the corresponding genes (Table 1).

**Table 1.** Forward and reverse primers for quantitative real-time PCR using SYBR Green for *Camptotheca acuminata* genes.

| Gene ID | Forward Primer Sequence (5′-3′) | Reverse Primer Sequence (5′-3′) |
|---------|--------------------------------|--------------------------------|
| CaUBC | CATCCAGAACCCGATAGTCC | TGTAAATTCACCCTTTCTTGG |
| CacGene03446 | AGAAGTGACGGGTTTTGTTTAG | CGGCCATTTGAAGAGAGATAC |
| CacGene06305 | TTGGGGAGGAGATGAAGAAG | AAGTTGGCTATGAGCGACCT |
| CacGene06306 | TGGAGTTTCAATGGACAATACC C | TTCAGAACAATAAGCCGCC |
| CacGene06505 | CCTCCAGTCTTTGATTGTT | ATTAGCAGAGCCCATTTT |
| CacGene08213 | CCCGAAACTGGACCTTGA | ATGGTCTCGTCCTCAGCG |
| CacGene15396 | TGGATGGTTGTCGGGAGT | TGTGGAAATTCTTGTGGC |
| CacGene14156 | GGGGAAGGCACAGGAAAT | GACCAACGGCACCAATCA |
| CacGene19114 | CCACCACTTCCAGACCAT | ACTCCAAATACCCATCCC |
| CacGene18366 | TCGCTTGGTGTCGGTATG | TAGGAGGAGCAGGTTGGT |
| CacGene17312 | GTAATCTTGTTGCCTCTTT | ACTTCCTGCTATCTCCTG |

## 3. Results

*C. acuminata* inoculated with the CaWB phytoplasma showed symptoms of leaf margin scorch from the bottom-up in the inoculated leaves at four weeks. At 10 WAI, the infected leaves began to scorch and fall off, many new shoots were drawn out, and the new apical shoots wilted. At 16 WAI, the new shoots were still accompanied by leaf margin scorch, the tree was weak, and a portion of *C. acuminata* died at 16 WAI, with a mortality rate of 22% (Table 2). At 20 WAI, the terminal buds were yellowed, internodes were shortened, and symptoms of leaf margin scorching had disappeared, and at 28 WAI, the internodes of the terminal leaves were shortened, and the characteristics of Witches'-broom appeared. This

indicates that the symptoms of *C. acuminata* are exacerbated with increasing inoculation time. Post-inoculation PCR revealed that the phytoplasma was detected in the terminal leaf at 2 WAI and was positive in the terminal leaf at all periods after 2 WAI, indicating that the CaWB phytoplasma could migrate into the apical leaf within two weeks (Figure 2).

**Table 2.** Mortality following *Camptotheca acuminata* infection.

| WAI | Starting Number of Individuals | Number of Survivors | Recovery Rate | Number of Deaths | Mortality Rate |
|---|---|---|---|---|---|
| X | N | Lx | Sx | Dx | Qx |
| 4 | 100 | 100 | 100% | 0 | 0% |
| 16 | 100 | 78 | 78% | 22 | 22% |

X: the time after *Camptotheca acuminata* has been infected, 4 WAI; N: Counts at the start; Lx: number of survivors; Sx: recovery rate = Lx/N; Dx: number of deaths; QX: mortality rate= Dx/N. WAI, weeks after infection.

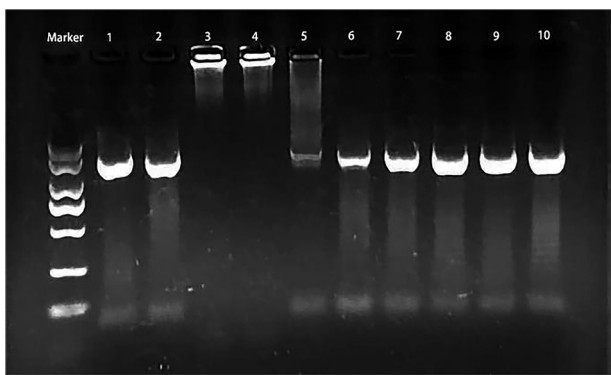

**Figure 2.** PCR analyses of the terminal tissues of *Camptotheca acuminata* after infection. M: marker, 1: CaWB tissues, 2: leafhopper (a vector of phytoplasma transmission), 3: healthy *Camptotheca acuminata*, 4–6: *Camptotheca acuminata* terminal tissues of 1–4 WAI; 7: 8 WAI tissue; 8: 16 WAI tissue; 9: 20 WAI tissue; 10: 28 WAI tissue. WAI, weeks after infection.

### 3.1. Transcriptome Differences in C. acuminata after Phytoplasma Infection

A set of 18 samples was collected from the terminal fifth leaves of *C. acuminata* plants at different stages of growth (4, 16, and 28 WAI) for sequencing. The sequencing process resulted in a total of 37.7–49.1 million clean reads, with an average of 45.6 million clean reads per library. In total, the transcriptome sequencing of *C. acuminata* yielded 123.11 Gb of clean reads with an average of 6.84 Gb of clean data per sample. The error rate observed during sequencing was <0.03%. The quality metrics Q20%, Q30%, and CG% were $\leq$98%, 95.2%, and 45.8%, respectively. A total of 37.7–47.2 million reads were successfully aligned with the reference genome [32], resulting in a matching rate of at least 81.95%. Subsequent analysis of the transcriptome data against the Swiss-Prot, Pfam, GO, and KEGG databases using the native BLAST algorithm revealed that of the 30,040 genes present in these databases, 21,612, 23,850, 13,747, and 14,124 genes were successfully annotated (Supplementary Tables S3–S5).

At 4 WAI, 2837 genes exhibited significant differences in expression between plants infected with CaWB and healthy plants, of which 1501 were significantly upregulated and 1336 were significantly downregulated. The criteria for significance were set at $|\log2(\text{FoldChange})|$ greater than or equal to 1 and a *p*adj value $\leq$ 0.05. Similarly, at the 16 WAI stage, 3682 DEGs were identified, of which 2,257 were upregulated and 1425 were downregulated (compared with healthy plants). At the 28 WAI stage, 3370 DEGs were identified, of which 2563 were upregulated and 807 were downregulated (compared with healthy plants) (Figure 3 and Table 3).

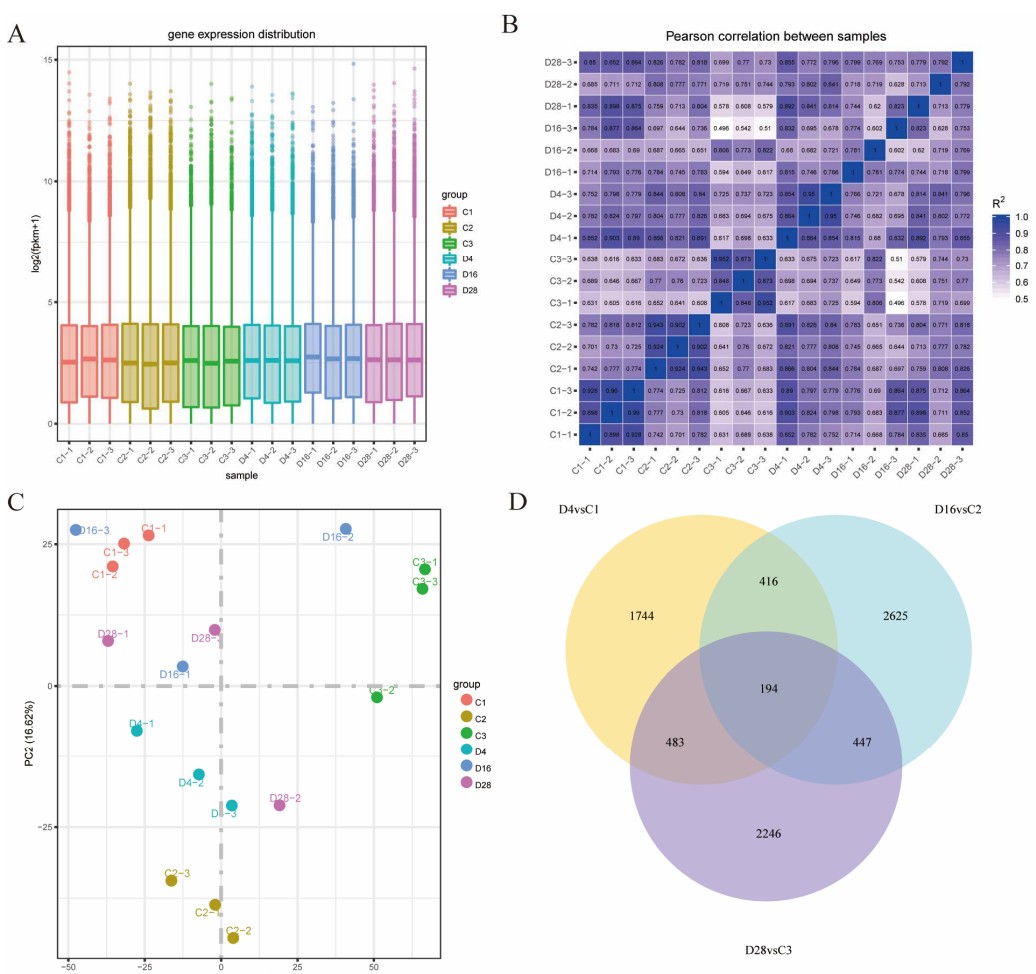

**Figure 3.** Gene expression in 4, 16, and 28 WAI diseased *Camptotheca acuminata* terminal leaves and respective controls. (**A**) Overall distribution of gene expression; (**B**) Pearson correlation coefficient between different treatments and their replicates; (**C**) principal component analysis; (**D**) Venn diagram of the differentially expressed genes where D4, D16, D28, C1, C2, and C3 represent 4 WAI, 16 WAI, 28 WAI, 4—week healthy leaves, 16—week healthy leaves, and 28—week healthy leaves, respectively. WAI, weeks after infection.

**Table 3.** Metabolites and mRNAs detected during phytoplasma infection of *Camptotheca acuminata*.

| Category | Metabolites | | | mRNAs | | |
|---|---|---|---|---|---|---|
| | 4 WAI | 16 WAI | 28 WAI | 4 WAI | 16 WAI | 28 WAI |
| Unique metabolites/gene detected | 989 | 989 | 989 | 27,063 | 27,149 | 27,032 |
| Significantly changed metabolites/genes | 133 | 105 | 126 | 2837 | 3682 | 3370 |
| Upregulated | 83 | 42 | 117 | 1501 | 2257 | 2563 |
| Downregulated | 50 | 63 | 9 | 1336 | 1425 | 807 |

### 3.2. GO Analysis of DEGs

At 4 WAI, the most prevalent category of DEGs was found to be related to molecular functions of biological processes. Specifically, 16 upregulated genes were associated with defense response, 42 with iron ion binding, and 5 with sucrose metabolic processes. Conversely, among the downregulated genes, 23 were involved in cell or subcellular movement, 23 were involved in microtubule motor activity, and 9 were involved in DNA-dependent DNA replication.

At 16 WAI, the majority of DEGs were also related to molecular functions. Among the upregulated genes, 84 were involved in hydrolase activity that specifically acted on

acid- and phosphorus-containing anhydrides. In addition, 82 genes were associated with pyrophosphatase activity. Furthermore, 72 upregulated genes were involved in hydrolase activity, acting on glycosyl bonds, 50 were associated with heme binding, and 18 downregulated genes were involved in oxidoreductase activity, specifically acting on paired donors with the incorporation or reduction of molecular oxygen (Supplementary Figure S1).

### 3.3. KEGG Analysis of DEGs

In transcriptome analysis, 14,124 genes were identified and annotated using the KEGG database. Among these genes, 1744 were specifically and significantly expressed only at 4 WAI. These genes were primarily enriched in pathways such as the ribosome, starch, and sucrose metabolism, plant hormone signal transduction, plant–pathogen interaction, and glycolysis/gluconeogenesis. Additionally, at 16 WAI, 2625 genes were specifically and significantly expressed, with the main enrichment in pathways related to carbon metabolism, biosynthesis of amino acids, photosynthesis, and endocytosis. Furthermore, at 28 WAI, 2246 genes were specifically and significantly expressed and primarily enriched in pathways such as plant hormone signal transduction, biosynthesis of cofactors, biosynthesis of amino acids, and ribosomes. The 194 DEGs at 4, 16, and 28 WAI were mainly enriched in pathways related to cutin, suberin, and wax biosynthesis; phenylalanine, tyrosine, and tryptophan biosynthesis; terpenoid backbone biosynthesis; starch and sucrose metabolism; and pentose and glucuronate interconversion (Figure 4).

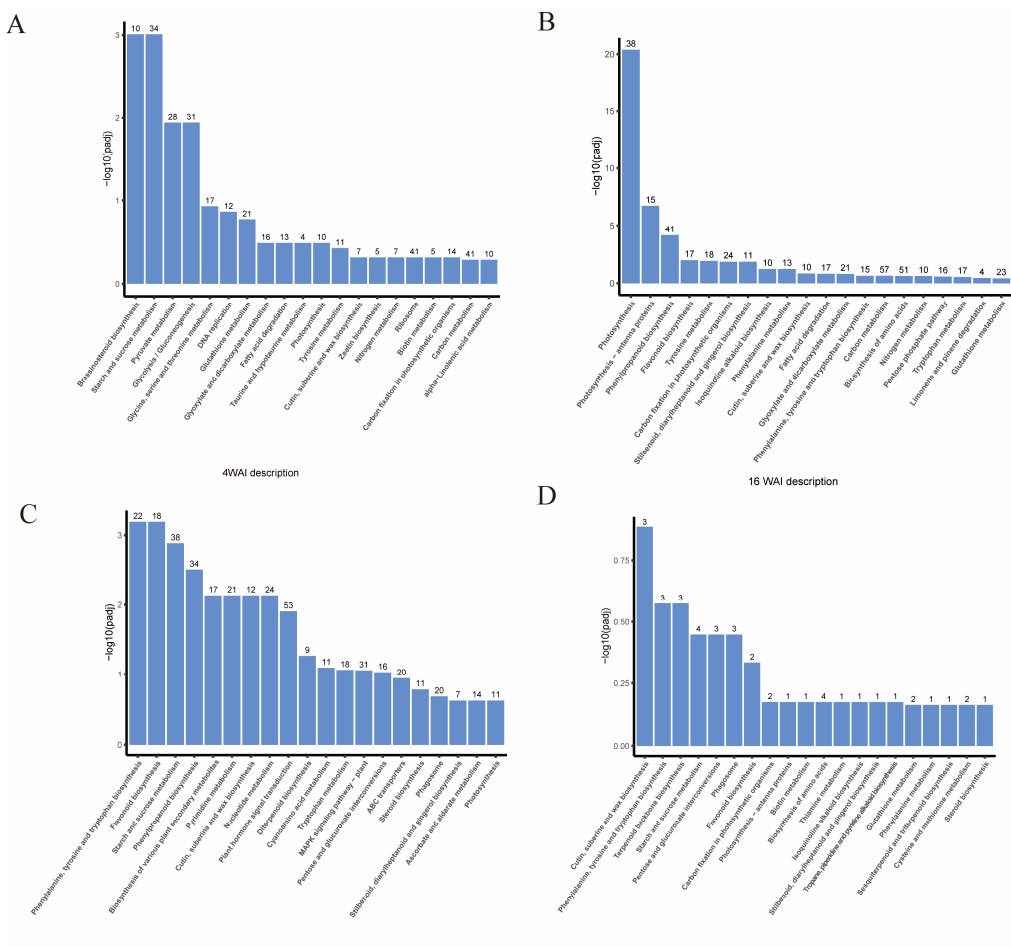

**Figure 4.** Top 20 enriched KEGG pathways based on DEGs in leaves of *Camptotheca acuminata* during CaWB phytoplasma infection. (**A**) 4 WAI infected vs. noninfected leaves; (**B**) 16 WAI infected vs. noninfected leaves; (**C**) 28 WAI infected vs. noninfected leaves; (**D**) 4, 16, and 28 WAI expressed the differential gene KEGG annotation. WAI, weeks after infection.

We identified the functional pathways specifically expressed by the plant and pathogen during different periods of infection from the DEGs of each WAI and the persistent signaling pathways of plant immunity to infection from the DEGs throughout the period of infection.

### 3.4. Differential Regulation of Pathways Involved in Carbohydrate Metabolism

Sucrose metabolism is important in various aspects of plant biology, including plant development, immune defense responses, and sugar metabolism. This metabolic process is influenced by the presence of phytoplasma in the vascular bundle and obstruction of sugar transport.

#### 3.4.1. Terpenoid Backbone Biosynthesis

At 4 WAI, sucrose phosphokinase (2.4.1.14), amylase (2.4.1.21), glucose-1-phosphate adenyltransferase (2.7.7.27), glycogen phosphorylase (2.4.1.1), alpha-amylase (3.2.1.1), and fructokinase (2.7.1.4) were upregulated, and beta-amylase (3.2.1.2), beta-glucosidase (3.2.1.21), and glucan endo-1,3-beta-D-glucosidase (3.2.1.39) were downregulated. At 16 WAI, sucrose phosphate kinase (2.4. 1.14), alpha-trehalose-phosphate synthase (2.4.1.15), alginate phosphatase (3.1.3.12), sucrose synthase (2.4.1.13), glucan endo-1,3-beta-D-glucosidase (3.2.1.39), beta-glucosidase (3.2.1.21), and cellulase (3.2.1.4) were upregulated, and glucose-1-phosphate adenyltransferase (2.7.7.27), glycogen phosphorylase (2.4.1.1), 4-alpha- glucanotransferase (2.4.1.25), and alpha-amylase (3.2.1.1) were downregulated. At 28 WAI, alpha, alpha-trehalose-phosphate synthase (2.4.1.15), trehalose-phosphatase (3.1.3.12), sucrose synthase (2.4.1.13), UDP glucose pyrophosphorylase (2.7.79), beta-fructofuranosidase (3.2.1.26), glucan endo-1,3-beta-D-glucosidase (3.2.1.39), beta-glucosidase (3.2.1.21), and cellulase (3.2.1.4) were upregulated, and sucrose phosphokinase (2.4.1.14), amylase (2.4.1.21), and alpha-amylase (3.2.1.1) were downregulated. In summary, *C. acuminata* undergoes complex changes in sugar metabolism with increasing infection, which are probably associated with changes in the photosynthetic system following phytoplasma infection, blockage of vascular bundles, and effects on sugar transport (Supplementary Table S6).

#### 3.4.2. Pentose and Glucuronate Interconversions

The pentose and glucuronate interconversion pathway is a central node of various biochemical reactions in organisms; hence, its regulation is involved in the production of various secondary metabolite precursors [33].

At 4 WAI, polyol dehydrogenase (1.1.1.14) and phosphoribulose epimerase (5.1.3.1) were upregulated, and phosphoribulose epimerase (1.1.1.22) was downregulated. At 16 WAI, phosphoribulose epimerase (1.1.1.22) was upregulated, whereas phosphoribulose epimerase (5.1.3.1) and pectate lyase (4.2.2.2) were downregulated. At 28 WAI, pectinesterase (3.1.1.11), pectate lyase (4.2.2.2), NADP-alcohol dehydrogenase (1.1.1.2), phosphoribulose epimerase (1.1.1.22), UDPG phosphorylase (2.7.7.9), and phosphoribulose epimerase (5.1.3.1) were upregulated, while no related enzyme-encoding genes were downregulated (Supplementary Table S6).

### 3.5. Differential Regulation of Signaling-Related Pathways

In healthy and diseased *C. acuminata*, the MAPK signaling pathway, plant-pathogen interaction, and phytohormone signaling pathways were the main signaling pathways that induced SAR in response to pathogen infection; we identified 72, 84, and 129 DEGs in these pathways, respectively (Supplementary Table S6).

#### 3.5.1. MAPK Signaling-Plant Pathway

In the MAPK signaling pathway, we found that flagellin sensing 2 (FLS2), a key gene in the pathogen infection pathway, was upregulated at 4 and 28 WAI, but not specifically at 16 WAI, probably because of the low titer of phytoplasma in new leaves at 16 WAI when old leaves were shed and new leaves were born. The expression of mitogen-activated protein kinase 3/6 (MPK3/6) was downregulated at 4 WAI and upregulated at

16 WAI. WRKY transcription factor 22/29 (WRKY22/29) was upregulated in all sample species. Pathogenesis-MAP kinase substrate 1 (MKS1) and WRKY transcription factor 33 (WRKY33) were upregulated at 16 WAI, and MKS1 was downregulated at 28 WAI. In the peroxide-induced pathogen attack pathway, serine/threonine-protein kinase (OXI1) was downregulated at 4 and 16 WAI and upregulated at 28 WAI. The expression of mitogen-activated protein kinase (ANP1) was downregulated at 4 WAI and upregulated at 28 WAI. WRKY22/29 was upregulated throughout the WAI and may be responsible for the dryness of the leaf margins due to peroxide-induced apoptosis. Mitogen-activated protein kinase kinase 17/18 (MAPK17/18) was downregulated at 4 WAI and upregulated at 16 and 28 WAI, while catalase isozyme 3 isoform X1 (CAT1) was upregulated at all WAI and may be involved with MAPK17/18 in the maintenance of ROS homeostasis.

### 3.5.2. Plant Hormone Signal Transduction Pathway

During phytoplasma infection of *C. acuminata*, all hormones are altered in response to the infection process. Among them, auxin transporter-like protein 4 (AUX1) was upregulated in all infected samples, and protein transport inhibitor response 1 (TIR1) was downregulated at 4 and 16 WAI and upregulated at 28 WAI. Auxin-responsive protein IAA (AUX/IAA) was downregulated at 4 WAI and upregulated at 28 WAI. Glycoside hydrolase 3 (GH3) and the auxin-responsive protein SAUR41-like (SAUR) were upregulated at 4 WAI. The transcripts associated with other genes in the same pathway exhibited both upregulation and downregulation. Specifically, in the cytokinin pathway, cytokinin response 1 (CRE1), a member of the two-component response regulator ARR-B family (BRR1), was upregulated at 4 WAI, whereas the type-A Arabidopsis response regulator (A-ARR) was downregulated at 16 WAI. In the abscisic acid pathway, pyrabatin resistance (PYR) and sucrose nonfermenting 1-related protein kinase 2 (SnRK2) were upregulated at 4 WAI. In the salicylate pathway, NPR1 was upregulated at 4 WAI and downregulated at 28 WAI, whereas TGA was downregulated at 4 WAI and upregulated at 16 WAI. Pathogenesis-related protein 1 (PR-1) was upregulated at 4 and 28 WAI, which may be correlated with enhanced disease-resistant secondary metabolism in plants during this period.

### 3.6. Differential Regulation of Secondary Metabolism Pathways
### 3.6.1. Phenylpropanoid and Flavonoid Biosynthesis Pathways

We compared different WAI of CaWB-infected *C. acuminata* with healthy controls and found that 189 DEGs were enriched in phenylpropanoid and flavonoid biosynthesis pathways, with the phenylpropanoid biosynthesis pathway being significantly affected. Notably, phenylalanine ammonia-lyase and ferulate-5-hydroxylase were differentially expressed throughout the WAI, whereas shikimate O-hydroxycinnamoyltransferase and coniferyl-alcohol glucosyltransferase were upregulated at 4 and 28 WAI. Caffeic acid 3-O-methyltransferase expression was downregulated at 4 WAI and upregulated at 16 WAI; trans-cinnamate 4-monooxygenase and 4-coumarate-CoA ligase in the phenylpropanoid synthesis pathway were upregulated during late infection at 28 WAI. Cinnamoyl-CoA reductase, 5-O-(4-coumaroyl)-D-quinate 3'-monooxygenase, caffeoyl shikimate esterase, cinnamyl-alcohol dehydrogenase caffeoyl-CoA O-methyltransferase, and coniferyl-aldehyde dehydrogenase were upregulated and may be highly correlated with the disease resistance response of *C. acuminata*. At 16 WAI, genes related to the flavonoid biosynthetic pathway were differentially expressed (upregulated) in large numbers, including chalcone isomerase, flavonoid 3'-monooxygenase, naringenin 3-dioxygenase, flavonol synthase, and leucoanthocyanidin reductase. Trans-cinnamate 4-monooxygenase, caffeoyl-CoA O-methyltransferase, chalcone synthase chalcone reductase, flavonoid 3'-monooxygenase, dihydroflavonol-4-reductase, naringenin 3-dioxygenase, and flavonol synthase/flavanone 3-hydroxylase were upregulated at 28 WAI and eucoanthocyanidin reductase was downregulated (Supplementary Table S6).



### 3.6.2. Flavone and Flavonol Biosynthesis

Flavones are disease-resistant secondary metabolites in plants that play a role in various pathogenic infections. Flavone and flavonol biosynthesis were altered at 16 WAI and 28 WAI, respectively, after phytoplasma infection of *C. acuminata*. Flavonoid 3′-monooxygenase [EC:1.14.14.82] regulates the synthesis of luteolin, quercetin, kaempferol 3-O-rhamnoside-7-O-glucoside, and kaempferol 3-O-beta-D-glucopyranoside. Quercetin 3-O-rhamnoside-7-O-glucoside synthesis by lavonol-3-O-L-rhamnoside-7-O-glucosyltransferase, and sophoraflavonoloside and baimaside synthesis by flavonol-3-O-glucoside/galactoside glucosyltransferase were upregulated (Supplementary Table S6).

### 3.7. Top DEGs with Upregulated/Downregulated Expression in Diseased C. acuminata

Supplementary Table S11 presents the top 10 genes that exhibited the greatest log2-fold change values in diseased leaves compared with their respective control samples. Among these genes, CacGene25418 demonstrated the highest log2-fold change value of 11.88 at 4 WAI.

The highest log2-fold change (11.88) was observed at 4 WAI for CacGene25418 (transcription factor ORG2), which was induced by OBP3, auxin, and salicylic acid (SA) in diseased plants compared to that in healthy plants. This was consistent with the KEGG pathway enrichment results, which showed that DEGs were also enriched by jasmonic acid (JA), UV light, and heat treatments. SA induces the expression of several disease-resistant proteins, including the stress-responsive A/B barrel domain (CacGene20866), which exhibits antibacterial activity against various pathogenic bacteria. Other genes specifically expressed at 4 WAI included the protein SRC2 homolog, protein NRT1/PTR FAMILY 5.7, monothiol glutaredoxin-S2, beta-glucosidase 24, and E3 ubiquitin-protein ligase. Among the downregulated genes at 4 WAI treatments, the DAG protein, chloroplastic (CacGene01678), acts very early in chloroplast development and is closely related to the post-infection leaf yellowing phenotype. Perakine reductase (*n* = 872) is an aldo-keto reductase involved in the biosynthesis of monoterpenoid indole alkaloids. Among the genes upregulated at 28 WAI, the nonspecific lipid transfer protein GPI-anchored 1 (CacGene07287) was the most differentially expressed gene. It is mainly associated with the lipid transfer protein LTPG2 that binds lipids and acts as a component of the epidermal lipid export mechanism, which extensively exports intracellular lipids (C29 alkanes) from epidermal cells to the surface to build keratin wax layers. Lipid-associated GDSL esterase/lipase At1g71691, probable glycerol-3-phosphate acyltransferase 3, and nonspecific lipid transfer protein were GPI-anchored 15. The GDSL esterase/lipase At1g29670 was also upregulated in the leaf tissues. The expression of the putative germin-like protein 2-1 and tropinone reductase-like 1 OS was also upregulated. During this period, cytochrome P450 71A9 (CacGene08939) was downregulated and presumably co-expressed with the corresponding transcription factor for ethylene, affecting leaf growth and blocking abscisic acid synthesis. This is consistent with the KEGG pathway enrichment results. The DEGs were also enriched in the plant hormone signal transduction pathway. At 28 WAI, the expression of protodermal factor 1 (a 14 kDa proline-rich protein), DC 2.15, and protein sodium potassium root defective 2, genes involved in terminal bud differentiation, were upregulated. In plants, phytoplasmas are usually distributed in vascular bundles, causing difficulties in sugar transport; herein, EP1-like glycoprotein 2 and jacalin-related lectin 3 were upregulated in plants involved in sugar metabolism. We found that *C. acuminata* disease resistance protein At4g27190 and reticuline oxidase were downregulated at 28 WAI during the fight against the pathogen, and secoisolar, which is involved in lignin synthesis, was downregulated at 28 WAI. Secoisolariciresinol dehydrogenase, which is involved in lignin synthesis, was also downregulated. Overall, *C. acuminata* expressed various disease-resistance genes during phytoplasma infection and showed disease tolerance at a later stage.

### 3.8. qRT-PCR Analysis of Selected Genes

To assess the reliability of our transcriptome sequencing data, we conducted quantitative reverse transcription PCR (qRT-PCR) on 10 DEGs involved in secondary metabolic pathways at 28 WAI. Both qRT-PCR and transcriptome sequencing analyses revealed consistent trends in the relative expression levels of all 10 DEGs (Figure 5). This observation supports the reliability of the gene expression changes identified by transcriptome sequencing analysis.

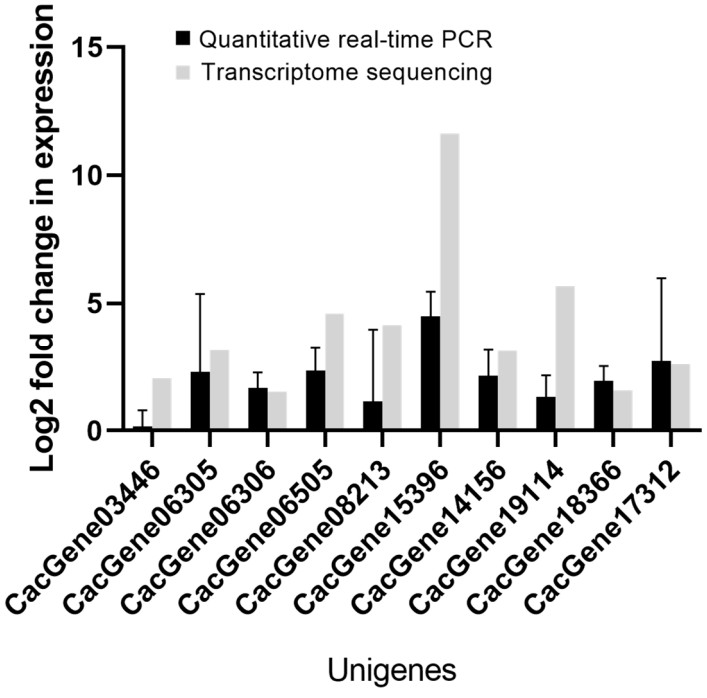

**Figure 5.** Relative expression of the 10 DEGs in the treated vs. control pairs validated using qRT-PCR at 28 WAI. The expression profiles of selected genes were determined using qRT-PCR (black) and transcriptome sequencing (gray) with the SYBR Green I chimeric fluorescence method, and the relative expression of each transcript was normalized using the CaUBC gene. The y-axis indicates the normalized expression level of the transcript, the x-axis indicates the gene, and the error bars indicate the standard deviation of the qRT-PCR signal.

### 3.9. Metabolomic Differences in C. acuminata after Phytoplasma Infection

HPLC–MS/MS analysis of the terminal fifth leaves at 4, 16, and 28 WAI identified 990 metabolites; PCA showed good quality among the biological replicates, with 23.79% and 16.51% variations in PC1 and PC2, respectively. At 4 WAI, *C. acuminata* had 133 differentially accumulated metabolites compared with healthy *C. acuminata*, of which 83 were upregulated and 50 were downregulated; 16 WAI *C. acuminata* had 105 differentially abundant metabolites, of which 42 were upregulated and 63 were downregulated; 28 WAI had 126 differentially abundant metabolites, with 117 upregulated and 9 downregulated metabolites. The PLS-DA model showed Q2Y values of 0.7, 0.63, and 0.74 for 4, 16, and 28 WAI, respectively. The overfitting validation results showed that all the samples had a Q2 value below zero, indicating that the model was not overfitted and could describe the samples well (Figure 6). As demonstrated by Venn diagrams, the three metabolites were differentially expressed throughout the WAI following phytoplasma infection, with 104 specific differentially abundant metabolites at 4 WAI, 83 specific differentially abundant metabolites at 16 WAI, and 104 specific differentially abundant metabolites at 28 WAI (Supplementary Tables S7–S9).

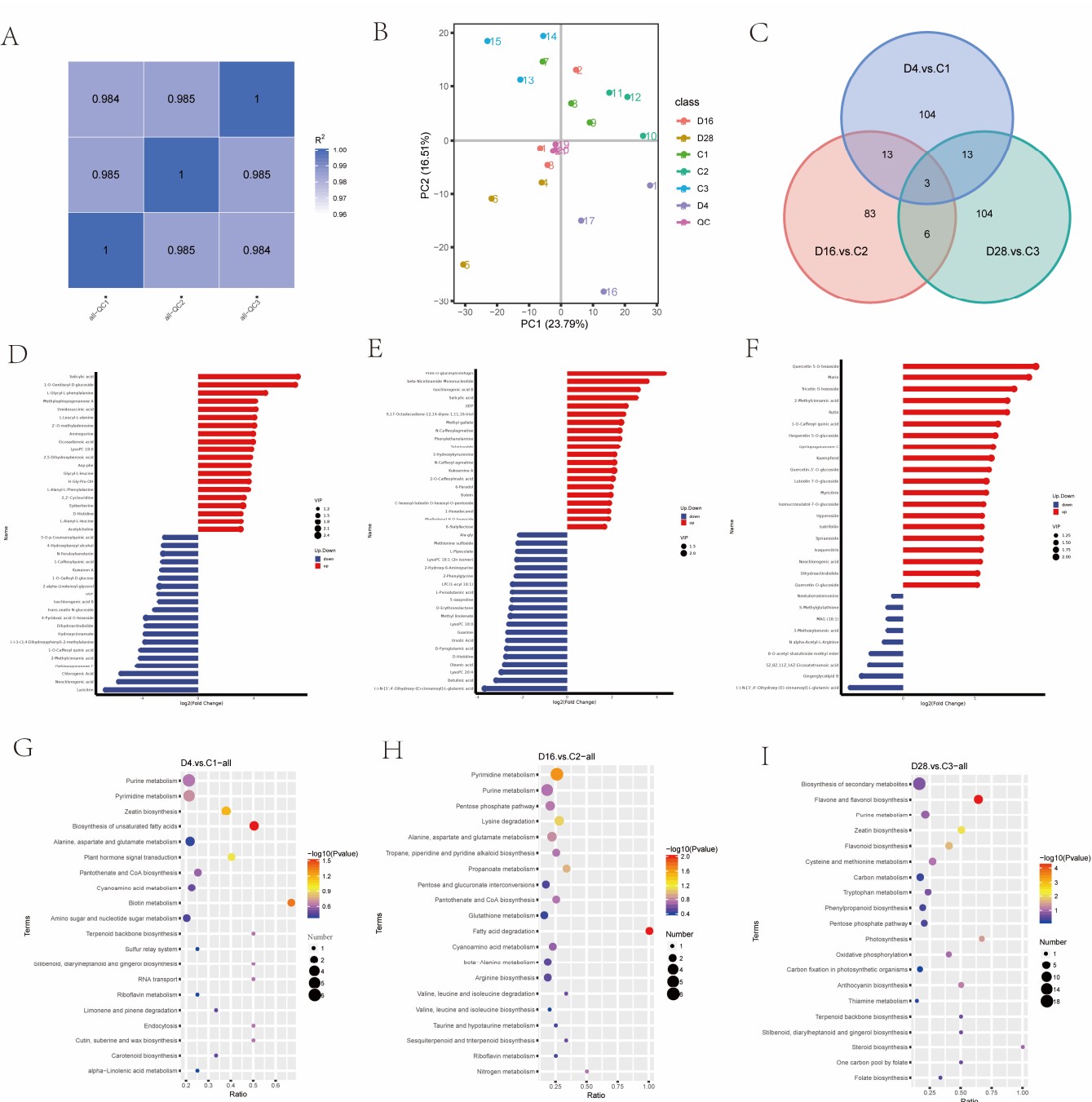

**Figure 6.** Summary of metabolomic analysis in 4, 16, and 28 WAI diseased *C. acuminata* terminal leaves and respective controls. (**A**) Pearson correlation coefficient between different treatments and their replicates; (**B**) principal component analysis; (**C**) Venn diagram of the detected metabolites; (**D**) top 10 (up- and downregulated) accumulated metabolites at 4 WAI; (**E**) top 10 (up- and downregulated) accumulated metabolites at 16 WAI; (**F**) top 10 (up- and downregulated) accumulated metabolites at 28 WAI; (**G**) scatter plot of KEGG pathways to which DAMs were enriched at 4 WAI; (**H**) scatter plot of KEGG pathways in which DAMs were enriched at 16 WAI; (**I**) scatter plot of KEGG pathways in which DAMs were enriched at 28 WAI. D4, D16, D28, C1, C2, and C3 denote 4 WAI, 16 WAI, 28 WAI, 4—week healthy leaf, 16—week healthy leaf, and 28—week healthy leaf, respectively. WAI, weeks after infection.

*3.10. KEGG Analysis of DAMs*

At 4 WAI, the primary pathways that exhibited enrichment were metabolic pathways with 28 substances and biosynthesis pathways of secondary metabolites with 17 substances. At 16 WAI, the main enriched pathways were metabolic pathways with 23 substances and purine metabolism with five substances. Finally, at the 28 WAI stage, the primary enrichment pathways were metabolic pathways with 27 substances and biosynthesis of secondary metabolites with 18 substances (Supplementary Table S10).

*3.11. Enrichment of Pathway-Specific DAMs in C. acuminata Affected by the Disease*

3.11.1. Plant Hormone Signal Transduction Pathway

SA level was significantly upregulated at 4 and 16 WAI. Gibberellin levels were upregulated and abscisic acid and cytokinin levels were significantly downregulated at 4 WAI. Ethylene level was significantly downregulated at 16 WAI. No significant differences in hormone levels were detected at the 28 WAI.

3.11.2. Biosynthesis of Secondary Metabolites

In this pathway, chlorogenic acid and caffeic acid levels were downregulated at 4 WAI and upregulated at 28 WAI, which may be correlated with the high expression of flavonoids at 28 WAI [34]. Gamma-dimethylallyl pyrophosphate and 5-aminoimidazole ribonucleotides accumulated significantly at 4 and 28 WAI, and the accumulation of 5-aminoimidazole ribonucleotides may be associated with nucleotide accumulation. SA and P-coumaryl alcohol significantly accumulated at 4 and 16 WAI, and the accumulation of SA may have contributed to changes in P-coumaryl alcohol. l-Ornithine, alpha-linolenic acid, histamine, and vitamin B2 were significantly accumulated at 4 WAI, whereas oleic acid, abscisic acid, O-acetylserine, (-)-carvone, and vanillin levels were significantly downregulated. At 28 WAI, s7p, ATP, isotrifoliin, 5-phosphoribosyl 1-pyrophosphate, 2-hydroxycinnamate, naringenin chalcone, rutin, adenosine 5′-diphosphate, and (-)-epigallocatechin were significantly accumulated. Chlorogenic acid, caffeic acid, and 5-aminoimidazole ribonucleotides may be involved in the regulation of these substances.

3.11.3. Purine Metabolism

Regarding purine metabolism, we found that ribose 1-phosphate level decreased at 4 WAI and was accumulated at 16 WAI. 5-aminoimidazole ribonucleotide level was observed as increased at 4 and 28 WAI. Additionally, guanosine 5′-diphosphate, ADP-ribose, 5-aminoimidazole ribonucleotide, and urea levels were upregulated and accumulated at 4 WAI, whereas dATP levels were downregulated. At 16 WAI, guanine and adenosine levels were significantly reduced. At 28 WAI, ATP, 5-phosphoribosyl 1-pyrophosphate, dGTP, and adenosine 5′-diphosphate were significantly accumulated.

3.11.4. Phenylpropanoid Biosynthesis Pathway

In this pathway, chlorogenic acid and caffeic acid levels were downregulated at 4 WAI, and P-coumaryl alcohol was significantly accumulated. P-coumaryl alcohol levels were downregulated at 16 WAI. Significant accumulation of chlorogenic acid, caffeic acid, and 2-hydroxycinnamate was observed at 28 WAI. The chlorogenic acid derivative, isochlorogenic acid B, decreased at 4 WAI but was significantly accumulated at 16 and 28 WAI.

3.11.5. Flavonoid Biosynthesis Pathway

In the flavonoid biosynthesis pathway, chlorogenic acid B levels decreased significantly at 4 WAI. Butein accumulated at 16 WAI, and the levels of chlorogenic acid, naringenin chalcone, kaempferol, and (-)-epigallocatechin were upregulated at 28 WAI. The high flavonoid expression at 28 WAI may be related to the survival strategy of *C. acuminata*.



## 4. Discussion

Plant–pathogen interactions induce a series of biological signal transduction that activate plant defense systems against pathogens [35,36]. There is extensive signaling interaction and recognition in triggering plant immune signaling networks [37]. Plant messenger signaling molecules, such as calcium ions, induce changes in cytoplasmic $Ca^{2+}$ concentration upon receipt of pathogen infection signals, triggering changes in a series of signaling molecules [38]. Compared with that in the control group, CDPK was upregulated and expressed at all three time points after *C. acuminata* was subjected to phytoplasma infection. This change could respond to the alteration in $Ca^{2+}$ levels in infected *C. acuminata* tissues as CDPK is regulated by $Ca^{2+}$ [39]. This upregulation might have led to the accumulation of ROS, thereby inducing an immune response including specific thickening of the cell wall [39]. In *Arabidopsis*, the activation of the bacterial secretion system results in KCS1/10 upregulation in all WAI. KCS1 is involved in very long-chain fatty acid synthesis in vegetative tissues and plays a role in wax biosynthesis [40]. The defense response gene, WRKY22/29, was also upregulated after infection, indicating a strong defense response to CaWB. At 4 WAI, plant hormone signaling molecules increase the expression of NPR1 and PR1 genes and decrease that of TGA genes. SA has also been reported to significantly accumulate during this period [41]. NPR1 binds to TGA induced by SA to regulate defense responses; PR1 is a marker of SA-mediated activation of systemic acquired resistance [42]. Therefore, we hypothesized that the SA-induced activated NPR1 binds to TGA and activates *C. acuminata* systemic resistance following inoculation. FLS2 is a model plant-surface receptor activated by bacterial flagellin flg22 to elicit a MAPK cascade response [43–45]. Phytoplasma may activate the MAPK signaling pathway via FLS2, thereby upregulating WRKY22/29. WRKY22/29, a nucleus-localized transcriptional activator, is involved in PTI and enhances resistance to pathogens in Arabidopsis, rice, and chili peppers, possibly due to the movement of phytoplasmas into the terminal leaflet, where they are captured by the FLS2 surface receptor; this receptor activates WRKY22/29 to initiate the plant defense response [46–48]. The reduced expression of these genes may be related to the regulation of oxidative hypersensitivity after necrosis in plant leaves.

In Arabidopsis, the nuclear substrate MKS1 regulates WRKY33 transcription factor activity downstream of innate immunity, thereby regulating plant antitoxin production [49,50]. At 16 WAI, WRKY33 showed increased expression, which may be due to the massive proliferation of phytoplasmas in the plant, thus initiating antitoxin production as a disease tolerance response to ensure survival. In jujube trees infected with jujube witches' broom, photosynthetic pigments and activity decreased, the expression of photosynthesis-related genes decreased, and leaf color turned yellow [51]. Infected *C. acuminata* had the highest number of differentially annotated genes for photosynthesis and accumulation of sugars during this period. This could explain why the phytoplasma clogged the vascular bundles, affecting sugar transport and causing plants to turn yellow from this period [52,53].

ANP1 is essential for cell division and the induction of immunity and development [54,55]. At 28 WAI, the ANP1 increased expression might have influenced the occurrence of Witches'-brooming. NDPK2 encodes nucleoside diphosphate kinase 2 in Arabidopsis and is a key gene in the response to growth hormones [56,57], while NDPK2 is a key gene in the response to growth hormones in *Camptotheca*. Changes in NDPK2 expression in *C. acuminata* may be associated with the occurrence of Witches'-broom. NPR1 is downregulated in plant hormone signal transduction; therefore, the upregulated expression of PR1 may be regulated by FLS2 [41,58]. We found a discrete dispersion between replicates within the group of diseased samples (D2 and D3) at 16 and 28 WAI in PCA, probably due to the different titers of phytoplasma in each plant at the time of inoculation and the fact that the symptoms of diseased *C. acuminata* exacerbate with time, with plants being weaker and the photosynthetic system suffering from the effects of phytoplasma at 16 and 28 WAI. Therefore, the individual samples in the treatment group presented by PCA-2D are more discrete but can still be divided into two categories from the control group.

Phenylpropanoid compounds play important roles in plant defense responses, such as cell wall thickening, by increased lignin or the generation of antimicrobial secondary metabolites for chemical defense responses [59–61]. Phenylpropanoids are also involved in inducing local or systemic immunity. The trends of DEGs and DAMs in the phenylpropanoid synthesis pathway found in *C. acuminata* are consistent with those reported in the date American palm cixiid (*Haplaxius crudus*) [22]. In particular, phenylalanine ammonia-lyase (PAL), shikimate O-hydroxycinnamoyltransferase (HCT), ferulate-5- hydroxylase (F5H), and coniferyl-alcohol glucosyltransferase were upregulated and expressed throughout the post-infection period, which may have led to the change of isochlorogenic acid B, atractylenolide II, and 3-methoxybenzoic acid in leaf tissues (Supplementary Table S10). The upregulation of PAL has been reported to activate broad-spectrum plant resistance [62], while that of HCT activates lignin synthesis and the binding of PR proteins to regulate plant defense responses [63]. As coniferyl-alcohol glucosyltransferase is a lignin precursor glycosyltransferase, the accumulation of phenylpropanoid analogs is expected [64,65]. We also observed that more genes were upregulated and expressed in the phenylpropanoid biosynthesis as the WAI increased, consistent with the trend of secondary metabolite accumulation within the leaf blades of diseased *C. acuminata*. Therefore, we conclude that an increase in phenylpropanoid compounds and their derivatives is a common response to plant disease resistance. However, these results vaguely describe the disease incidence pattern of *C. acuminata*, and the mechanisms of regulation of the defense response in each WAI remain to be investigated. Since this study's material was grown in a laboratory greenhouse, the climatic conditions may differ from the natural environment and *C. acuminata* was inoculated with phytoplasma only once. Frequent feeding on *C. acuminata* by vector insects under natural conditions may exacerbate phytoplasma infection, and climatic conditions may influence *C. acuminata* resistance mechanisms. Therefore, the present study had some limitations, and in the future, we could design experiments in plantations with established diseases to obtain more reliable insights.

## 5. Conclusions

The current study demonstrated the complex signaling network regulation and changes in secondary metabolites in the disease incidence pattern of *C. acuminata*. In particular, the plant–pathogen interaction pathway significantly increased the expression of CDPK, KCS1/10, and WRKY22/29, which are possible key genes for the defense response to CaWB and the maintenance of ROS homeostasis. We also found that isochlorogenic acid B, atractylenolide II, and 3-methoxybenzoic acid were differentially enriched throughout the post-inoculation period, which may signal a defense response or the presence of functional substances. Future studies should identify potential candidates to enhance plant disease tolerance by evaluating the survival rate of plants subjected to lethal infections after the exogenous spraying of candidate substances. The findings of this study provide novel insights into the interactions between phytoplasmas and *C. acuminata* and the defense response mechanisms of plants subjected to phytoplasma infections.

**Supplementary Materials:** The following supporting information can be downloaded at https://www. mdpi.com/article/10.3390/agriculture13101943/s1. Additional file S1: Supplementary Table S1. Summary of transcriptome sequencing of infected and noninfected *C. acuminata* tissues. Supplementary Table S2. Summary of reads and reference genome comparison statistics. Supplementary Table S3. Differential expression of genes between infected and noninfected *C. acuminata* at 4 WAI. Supplementary Table S4. Differential expression of genes between infected and noninfected *C. acuminata* at 16 WAI. Supplementary Table S5. Differential expression of genes between infected and noninfected *C. acuminata* at 28 WAI. Supplementary Table S6. List of differentially expressed genes enriched for specific pathways in different WAI of terminal new leaves of CaWB-infected *C. acuminata* compared with controls. Supplementary Table S7. Accumulation of different metabolites between infected and uninfected *C. Acuminata* at 4 WAI. Supplementary Table S8. Accumulation of different metabolites between infected and uninfected *C. acuminata* at 16 WAI. Supplementary Table S9. Accumulation of different metabolites between infected and uninfected *C. acuminata* at 28 WAI. Supplementary

Table S10. Pathway-specific differential accumulation of metabolites in different WAI of terminal new leaves of infected *C. acuminata* compared with noninfected tissues. Supplementary Table S11. Top DEGs with increased/decreased expression in diseased *C. acuminata*. Additional file S2: Supplementary Figure S1. Histogram showing GO annotation of the differentially expressed genes. Supplementary Figure S2. Symptoms of *C. acuminata* terminal leaves or infection leaves at different stages of infection by CaWB. (A) 4 weeks after infesting (WAI); (B) 10 WAI; (C) 16 WAI; (D) 20 WAI; (E) 28 WAI; WAI, Week after infection.

**Author Contributions:** Conceptualization, H.C.; methodology, K.Q.; software, W.H.; validation, K.Q., X.L. and W.H.; formal analysis, K.Q., J.L., X.L. and W.H.; resources, H.C.; data curation, K.Q.; writing original draft preparation, K.Q. and H.C.; writing review and editing, K.Q. and H.C.; visualization, W.H., J.L. and K.Q.; supervision, X.L., W.H. and J.L.; project administration, H.C.; funding acquisition, H.C. All authors have read and agreed to the published version of the manuscript.

**Funding:** This study was supported by the Regional Science Foundation Program of the National Natural Science Foundation of China (31960535).

**Data Availability Statement:** Data supporting the findings of this work are provided within the paper and its Supplementary Information files. A reporting summary for this article is available as a Supplementary Information file. The data set and plant materials generated and analyzed during the current study are available from the corresponding author upon request. The raw transcriptome data have been submitted to NCBI SRA under the project number PRJNA960904 (PRJNA960904 Details | Manage Data | Submission Portal (nih.gov (accessed on 14 April 2023)). The reference genome used in this study is from Kang et al. published article, chromosome-scale genome assembly (https://doi.org/10.6084/m9.figshare.12570599 (accessed on 14 April 2023)) and the GFF3 file (https://doi.org/10.6084/m9.figshare.12570614 (accessed on 14 April 2023)) are available in Figshare. The SwissProt and Pfam databases used in this study are available at https://www.uniprot.org (accessed on 14 April 2023) and Pfam: Home page (xfam.org (accessed on 14 April 2023)). The KEGG pathway database is available at https://www.kegg.jp (accessed on 14 April 2023). The GO database is available at Gene Ontology Resource. Source data are provided in this paper.

**Acknowledgments:** This work was completed in the Phytopathology Laboratory of Yunnan Province. We thank Novogene Co., Ltd. for providing transcriptome sequencing, metabolome sequencing, and data analysis.

**Conflicts of Interest:** The authors declare no conflict of interest. The funders had no role in the design of the study; in the collection, analyses, or interpretation of data; in the writing of the manuscript; or in the decision to publish the results.

## Abbreviations

CaWB: *Camptotheca acuminata* witches'-broom phytoplasma disease; WAI: week after the inoculation; DEG: Differential Expression Analysis; MAPK: mitogen-activated protein kinases; SAR: systemic acquired resistance; FLS2: flagellin sensing 2; MPK3/6: mitogen-activated protein kinase 3/6; MKS1: Meckel–Gruber syndrome; OXI1: Serine/threonine-protein kinase 1; MAPK17/18: mitogen-activated protein kinase 17/18; CAT1: catalase isozyme 3 isoform X1; CDPK: Calcium-dependent protein kinase; DAM: Differentially accumulated metabolites; MPK3/6: Mitogen-activated protein kinase 3/6; PTI: Plant Innate Immunity; CsLOB1: Citrus sinensis lateral organ boundary 1; NDPK2: Nucleoside diphosphate kinase; TGA: TFTGA; NPR1: nonexpressor of pathogenesis-related genes 1; ABA: Abscisic Acid; CTK: cytokinin; ROS: reactive oxygen species; KCS1/10: β Ketoacyl-CoA Synthase1/10; PAL: phenylalanine ammonia-lyase; HCT: shikimate O-hydroxycinnamoyltransferase; F5H: ferulate-5-hydroxylase; CNGCs: cyclic nucleotide gated channel; CAM/CML: calcium-binding protein CML; FLS2: flagellin sensing 2; PR-1: pathogenesis-related protein 1; TIR1: Transport Inhibitor Response 1; AUX/IAA: auxin-responsive protein IAA; GH3: Glycoside hydrolase 3; SAUR: Small Auxin Up-Regulated genes; CRE1: cytokinin response 1; BRR1: ARR-B family; A-ARR: Type-A Arabidopsis response regulator; PYR: Phosphoenolpyruvate; SnRK2: Sucrose nonfermenting 1-related protein kinases 2; SA: salicylic acid; JA: jasmonic acid; DAG: Diacylglycerol; GPI-anchored 1: Glycosylphosphatidylinisotol anchored 1; and LTPG2: Lipid Transfer Protein 2.

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
