# Peer review of "Combined Transcriptomic and Metabolomic Analyses of Defense Mechanisms against Phytoplasma Infection in Camptotheca acuminata Decne"

_agriculture, doi:10.3390/agriculture13101943_

Round 1

Reviewer 1 Report

Authors have presented their work on title: Combined transcriptomic and metabolomic analyses of defense mechanisms against phytoplasma infestation in Camptotheca acuminata Decne.

There are some minor corrections:

Abstract: There should be specific data presented in the abstract, it is very generic.

Introduction: It is quite a surprise that authors had to use a reference from 1966. Please update ref #2.

M&M: Poor writing of this section, the WAI are not clearly described (page 3, lines 99-104 & 125 - 133) What was the tissue used? 

Results: Check all the figure legends for the spellings and font type, font type should be uniform. 

Discussion: Good discussion

Conclusion: Very generic conclusion, could have been more specific in light of findings reported.

Please check for the grammatical and spelling mistakes.

Reviewer 2 Report

The topic under study “Combined transcriptomic and metabolomic analyses of defense mechanisms against phytoplasma infestation in Camptotheca acuminata Decne.” is very interesting and current. The manuscript is clearly written, well-structured, and the results and discussion are also adequate.

Minor corrections:

Line 13 – “…CaWB. phytoplasma ….”

35 – erase full stop “effects.[1,2]”

Line 52 – I suggest changing “role in plant-host interactions” to “role in plant host interactions” or “role in plant-pathogen interactions”

Line 76 – add full stop “5 days Afterward”

Line 242 – space missing “or equal to 0.05.Similarly, at the”

Line 246 – erase full stop “to healthy plants. (Figure”

Line 271 - erase full stop “molecular oxygen. (Sup-271 plementary”

Line 313 – correct “at 28 WAI) alpha, alpha-trehalose”

Line 375 – erase 1 full stop “particular period.. In the salicylate”

Line 389 – correct “upregulated at 16 WAI. trans-cinnamate”

Line 413 - check endpoint (also in lines: 502, 565, 689, etc)

Line 465 – correct “acuminata Differentially”

Line 478 – insert a space “well(Figure 6.).”

Line 505 – correct “infestation. gibberellins were”

Line 506 – correct “at 4 WAI. ethylenes”

Line 520 – correct “accumulated. chlorogenic”

Line 538 – write Favonoid “3.13.5. favonoid biosynthesis pathway”

Line 540 – correct “4 WAI.Butein significantly”

Line 589 – correct “grown leaves.[33]. WRKY3”

Line 598 - write the names in Latin in italics throughout the document

658 – correct “were upregulated.[26”

- The main question addressed by the research is clearly defined in lines 64 to 68.

- The topic covered "Plant resistance mechanisms" has the greatest importance and relevance for improving the sustainability of production systems.

- The studied host-pathogen interaction is important in the region, which does not take away its relevance in terms of research.

- The results presented are innovative.

- The methodology used is frequently followed in studies on host-parasite interactions and the conclusions seem appropriate to the results obtained.

- Finally, the references are current.

Reviewer 3 Report

In this paper, changes in gene expression in Camptotheca acuminata plants following infections by Camptotheca acuminata witches’-broom (CaWB) phytoplasma were examined through transcriptomic and metabolomic analyses at three different times (2, 16 and 28 weeks after inoculation [WAI], respectively) during the disease development. A major concern with this paper is that authors’ deductions and conclusions are not scientifically sound and justified by the data presented. In addition, the paper is not well written and there are some flaws in the experimental design.

Specific criticisms:

Throughout the paper. Both wording and terminology are often incorrect which make it sometimes difficult to understand what the authors mean. For instance, the term “phytoplasma infestations” is erroneously used throughout the text in place of “phytoplasma infections”. Very often “infestation” is used istead of “inoculation”. Authors give the impression that are not aware of the several stages of the disease development. What does “infestation cycles” mean? Do authors consider CaWB disease a polyciclic disease?

Page 2, lines 35-39. More appropriate references about the occurrence of CaWB disease in China should be provided (e.g., Wang et al, Acta Phytopathologica Sinica, 2021, 51(3): 429-440).

Page 2, line 40. About the statement “Phytoplasmas are significant prokaryotic pathogens that form biofilms”, please provide firm evidence (reliable references) that this is truee.

Page 3, lines 64-68. This text seems a “cut and paste” from another paper. In this paper P. hippocastanum was not examined.

Page 3, lines 70-86. This part of “Material and Methods” section is not clear and should be re-written. Usually leafhoppers employed in transmission experiments are reared on phytoplasma-infected plants to acquire phytoplasma infections but are not incubated with infected plants. Also, the text “The seedlings were incubated on healthy Pleasant Tree plants for a period of 5 days Afterward, they were transferred to healthy Camptotheca acuminata plants for a 3-day inoculation period” need clarification. This sentence “the older leaves showed signs of water staining” is not correct. “Signs” should be replaced by symptoms and “water staining” by water-soaked areas or water-soaked spots.

Pages 3-4. “DNA extraction and PCR analysis” section. In this section there is no mention that PCR assays were conducted also from DNA extracted from leafhoppers whereas Figure 2 of the “Results” section shows PCR products from leafhopper sample. CaWB phytoplasma is a member of the phytoplasma taxonomic group 16SrXXXII. However data on the taxonomic identity of phytoplasma(s) detected in Camptotheca acuminata plants are not provided since authors used for PCR assays only universal phytoplasma primers. These data can be obtained through RFLP and/or sequence analyses or specific primers.

Throughout the paper. Authors should explain why they used terminal leaves for their investigations without elucidating the colonization behavior of CaWB phytoplasma in Camptotheca acuminata plants.

Page 6, line 202. “virulent leafhoppers” should be replaced by “inoculative or infective leafhopper” since CaWB phytoplasma is not a virus.

Page 6, lines 203-213. Figure 2 does not show clear cut phytoplasma symptoms (e.g., witches’-broom) as claimed in the text. Also, leaf scorching and water-soaked areas (shown in panel b) are not typical symptoms.

Throughout the paper. Authors claimed that symptom severity increased over time during the disease development: However, they did not relate this finding to the profound changes in gene expression, suggestive of disease resistance, occurring in the late stage of the disease development. This issue should be properly addressed.

Due to problems described, I do not think the manuscript meet criteria for publication in Agriculture.

English should be improved.

Round 2

Reviewer 3 Report

Major reviewer’s concerns are not satisfactorily addressed in the revised version of this paper.

Wording and terminology are often incorrect.
